# Say their names: Resurgence in the collective attention toward Black victims of fatal police violence following the death of George Floyd

Henry H. Wu[1,2,4], Ryan J. Gallagher[5], Thayer Alshaabi[6], Jane L. Adams[7], Joshua R. Minot[1,2,3], Michael V. Arnold[1,2,3], Brooke Foucault Welles[5], Randall Harp[2,8], Peter Sheridan Dodds[1,2,3,4,9,10], Christopher M. Danforth[1,2,3,4,11]*

1 Computational Story Lab, Burlington, VT, United States of America, 2 MassMutual Center of Excellence for Complex Systems & Data Science, Vermont Complex Systems Center, Burlington, VT, United States of America, 3 Vermont Advanced Computing Center, University of Vermont, Burlington, VT, United States of America, 4 Harvard University, Boston, MA, United States of America, 5 Communication Media & Marginalization Lab, Network Science Institute, Northeastern University, Boston, MA, United States of America, 6 Advanced Bioimaging Center, University of California, Berkeley, CA, United States of America, 7 Department of Computer Science, Northeastern University, Boston, MA, United States of America, 8 Department of Philosophy, University of Vermont, Burlington, VT, United States of America, 9 Department of Computer Science, University of Vermont, Burlington, VT, United States of America, 10 Santa Fe Institute, Santa Fe, NM, United States of America, 11 Department of Mathematics & Statistics University of Vermont, Burlington, VT, United States of America

* chris.danforth@uvm.edu

## Abstract

The murder of George Floyd by police in May 2020 sparked international protests and brought unparalleled levels of attention to the Black Lives Matter movement. As we show, his death set record levels of activity and amplification on Twitter, prompted the saddest day in the platform's history, and caused his name to appear among the ten most frequently used phrases in a day, where he is the only individual to have ever received that level of attention who was not known to the public earlier that same week. Importantly, we find that the Black Lives Matter movement's rhetorical strategy to connect and repeat the names of past Black victims of police violence—foregrounding racial injustice as an ongoing pattern rather than a singular event—was exceptionally effective following George Floyd's death: attention given to him extended to over 185 prior Black victims, more than other past moments in the movement's history. We contextualize this rising tide of attention among 12 years of racial justice activism on Twitter, demonstrating how activists and allies have used attention and amplification as a recurring tactic to lift and memorialize the names of Black victims of police violence. Our results show how the Black Lives Matter movement uses social media to center past instances of police violence at an unprecedented scale and speed, while still advancing the racial justice movement's longstanding goal to "say their names."

**Data Availability Statement:** "***PA@ACCEPT: Request SI data file*** The data from this study was downloaded from a public database Fatal

Encounters, and a public API at http://storywrangling.org which provides counts for phrases on Twitter. Our collection and analysis of the data in this study complies with the terms and conditions of the data sources. Individuals interested in reproducing the results of the study can retrieve the data from these public sources.

**Funding:** PSD and CMD received support from a gift to the UVM Foundation from MassMutual "The funders had no role in study design, data collection and analysis, decision to publish, or preparation of the manuscript."

**Competing interests:** The authors have declared that no competing interests exist.

# 1 Introduction

On February 23rd, 2020, Ahmaud Arbery, a 25-year-old Black man, was shot and killed by three white men while jogging in Georgia. On March 13th, Breonna Taylor, a 26-year-old Black woman, was fatally shot in the crossfire of a "no-knock" apartment search by police in Kentucky. And on May 25th, George Floyd, a 46-year-old Black man, was arrested outside a convenience store in Minnesota, and murdered when white police officer Derek Chauvin knelt on his neck for over 9 minutes. Catalyzing the anger and grief that had been circulating the deaths of Arbery and Taylor, Floyd's murder sparked protests across the United States, bringing renewed attention to police brutality and racism. The protests were coupled with an unprecedented use of the hashtag #BlackLivesMatter [1], surpassing every previous surge of the hashtag since its introduction in July 2013 following the acquittal of George Zimmerman in the death of Trayvon Martin and its widespread adoption in November 2014 following the non-indictment of police officer Darren Wilson in the death of Michael Brown [2, 3]. It is estimated that 15 to 26 million Americans participated in racial justice protests in June 2020, making them the largest protests in American history [4].

Continuing the work of a centuries-long racial justice movement, Black Lives Matter has connected many individual instances of police violence against Black individuals into a larger narrative about systemic racism in the United States. In December 2014, the African American Policy Forum and the Center for Intersectionality and Social Policy Studies launched the #SayHerName campaign, which "brings awareness to the often invisible names and stories of Black women and girls who have been victimized by racist police violence." [5] Since then, the related phrase "say their names" and the hashtag #SayTheirNames have been invoked to recognize Black victims of police violence more broadly. While some have critiqued #SayTheirNames for drawing attention away from the women and girls that are the focus of #SayHerName, repeating the names of victims of police violence of all genders serves an important narrative function. Naming victims memorializes and celebrates these individuals, while emphasizing their place in a larger system of police violence and racial prejudice [6]. This rhetorical strategy contributes to the goals of Black Lives Matter activists and the racial justice movement more broadly, which include "education, amplification of marginalized voices, and structural police reform" [3].

Here, we first characterize the wave of attention that was given on Twitter to the death of George Floyd and the subsequent protests. His death prompted an unparalleled surge in tweet volume driven by historic levels of retweet amplification, which coincided with a substantial and sustained dip in the happiness expressed on the platform. Of the few dozen individuals who have ever received comparable amounts of attention online, George Floyd is the only person who was not already a celebrity. By using the names of 3,737 prior Black victims of police violence to contextualize how George Floyd's name was said alongside others, we find that the protests surrounding his death brought a resurgence in attention to 186 past instances of fatal police violence against Black Americans. This resurgence was instantaneous and more persistent than previous spikes in attention to the Black Lives Matter movement. We use the connection between George Floyd and past Black victims to place the particular viral moment of his death alongside the many other moments of heightened attention to police violence over the past 12 years on Twitter. Across the past decade, we see patterns of increasing attention to and amplification of Black victims of fatal police violence and discernible trends in how often and widely different names have been used. Our study goes beyond the most emblematic names of the Black Lives Matter movement and shows how social media provides activists and allies the opportunity to connect thousands of less visible—but no less important—instances of racial police violence together at a scale and speed that was not previously possible.

## 2 Related work

### 2.1 Bearing witness through the Black public sphere

The Black Lives Matter movement and its corresponding hashtag #BlackLivesMatter are so impactful because they build on a long history of Black people bearing witness to anti-Black violence perpetrated by slavery, lynching, and police brutality [7]. Accounts of anti-Black violence prior to the 20th century were passed from person to person in communal spaces like churches [8, 9]. Not only did this add to the established collective memory of brutality enacted by slavery [10], but it further allowed Black people to affirm one another's experiences of violence and oppression and strategize ways to protest against them [8]. Those stories and experiences were transcribed by early members of the Black press, including Samuel Cornish and John Russwurm, who founded the first Black newspaper in the United States, *Freedom's Journal*; abolitionist Frederick Douglass, who founded the *North Star*; and journalist Ida B. Wells, who extensively documented lynching in the South [7, 8]. In bearing witness to and drawing connections across events of anti-Black violence, they memorialized Black victims by showing their deaths were "thematic, rather than episodic" instances of anti-Black racism [7].

The writings published by the Black press challenged the racist narratives espoused by the mainstream—and overwhelmingly white—public sphere [11]. By dismantling white supremacist discourses, the Black press helped the Black public sphere emerge: a *counterpublic* sphere organized around texts, songs, stories, radio shows, and everyday talk that speak to Black experiences, including those of bearing witness to anti-Black violence [12–14]. The Black counterpublic externally advocates against the racist ideologies of the mainstream public sphere, while internally affirming experiences of anti-Black violence and providing discursive refuge from white supremacy [15]. By encompassing the writings of the Black press and all other writings bearing witness to anti-Black violence, the Black public sphere symbolically unifies the collective memory of violence against Black communities.

Historically, the Black counterpublic sphere relied on flyers, pamphlets, word-of-mouth, and demonstrations to capture the attention of white communities—a necessary causal factor in producing sustainable, structural change—and make them confront the realities of anti-Black violence. This already difficult task was not made easier by the growth in print and television news media through the 20th century, which bottlenecked access to white audiences through media gatekeepers that barred antiracist messaging [16, 17]. In the past and still, this binds racial justice advocates: in order to direct attention to particular racial justice campaigns, activists are forced to publicly and disruptively protest. While this can attract the attention of mainstream news—and, therefore, white audiences—it comes at the cost of the "protest paradigm," the strikingly consistent way in which mainstream media attempts to delegitimize protests because of their disruptions to public spaces [18–20]. As evidenced by multiple racial justice campaigns, most notably the Civil Rights movement of the 1950s and 1960s, antiracist policies and ideas can still gain traction in the face of the protest paradigm. However, it is a persistent challenge for racial justice campaigns to both frame their own narratives and have them widely broadcasted when that access is regulated through mass broadcast media.

### 2.2 Racial justice hashtag activism

The introduction of the internet dramatically restructured the possibilities for Black communities and allies to challenge racist narratives in the mainstream public sphere. Rather than reserving the means of producing and disseminating information en masse for a small set of journalists, political elites, and celebrities, the internet makes it easy for everyday people to connect with one another and quickly disseminate information to massive audiences [21, 22].

Black people, in particular, were innovative early adopters of the internet [23] and took to social media platforms like Twitter at higher rates than others in the United States [24]. The technological fluency of "Black Twitter" and its distinct discursive style merged well with the platform's fast pace and hashtag-organized interface [25, 26]. In particular, Black Twitter's ability to coalesce around trending topics and emerging news stories through the use of hashtags gives it the capacity to act like an "ad hoc news outlet that breaks news and supplies updates in real-time, rivaling some of the most time-honored legacy media" [7]. Exactly for that reason, journalists look towards Black Twitter and other online communities to understand emerging events and people's reactions to them [27, 28]. Rather than having to appeal directly to print and news media, activists and others can find their messaging distributed more broadly in the public sphere when it is picked up by journalists via social media [29, 30].

When those emerging news stories concern police violence against Black victims, the networks of Black people and their allies online function as a *networked* counterpublic that drives antiracist narratives into the mainstream conversation [31]. Like the historically offline Black public sphere, the goal of networked counterpublics is to advance racial justice by reframing the dominant discourse and engaging sympathetic new audiences [32, 33]. Unlike the offline counterpublic sphere, though, networked counterpublics have the technical tools to engage in *connective* action that string together experiences of oppression and marginalization in real time at a scale that was not previously not possible [34, 35]. In particular, hashtags—which are used particularly effectively by Black Twitter [25]—provide a common banner under which people can share their stories, affirm the experiences of others, and challenge racist mainstream discourses [6]. Rather than replacing the offline functions of the Black public sphere [15], networked counterpublics' ability to digitally weave many individual experiences into a larger collective tapestry of oppression [34, 36] makes it possible to bear witness to anti-Black violence in new ways that emphasize the scale of the injustice.

## 2.3 Bearing witness through #BlackLivesMatter

In these ways, the Black Lives Matter movement is the culmination of the Black public sphere's memorialization of Black victims of anti-Black violence. It should not be viewed as a replacement of offline advocacy, but as another tactic for bearing witness to anti-Black violence working in tandem with on-the-ground organizing, particularly since hashtag activism bears its own difficulties. Connective action is not inevitable from the introduction of a hashtag: the vast majority of instances of online activism never reach viral levels of attention [37]. Posts and hashtags about racial injustice compete with thousands of other topics at any given time, and they can easily fall through the emergent process of what is amplified and what is not if a critical mass is not achieved [38, 39]. Further, the competition for attention takes place on platforms that do not have neutral values or algorithmic curations [40–42]. Anti-Black algorithmic bias regularly appears across online platforms in terms of search, visibility, and amplification [43]. Together, the convergence of these factors suppress many instances of racial injustice and limits the ability of networked counterpublics to emerge around them.

Yet, the Black Lives Matter movement materialized despite these barriers. Moreover, it has persisted across several years, unlike many other instances of online activism [44, 45]. While this is due in no small part to offline organizing for the movement, it also due to how #BlackLivesMatter has become more than just a tag in posts: it is a *signifier* of a broader idea, feeling, and movement [35]. The hashtag #BlackLivesMatter, for example, does not just stand for the death of Trayvon Martin, for whom it was originally invented, nor Michael Brown, for whom the hashtag gained widespread visibility. It signifies a recognition of all past and future Black victims of police violence; it signifies a commitment to addressing violence against Black

communities more broadly; and it signifies attention to persistent and historical systemic racism, particularly in the United States.

This symbolic significance is enshrined through the phrases "say her name" and "say their names" that have become emblematic of the Black Lives Matter movement. They both explicitly encourage people to say, repeat, and amplify the names of Black victims of fatal police violence, an important discursive strategy of the movement [3, 6, 46–48]. Even though each new instance of extrajudicial police violence is very often not *directly* related to a past incident, the names of past victims are still often reiterated online in the wake of a new victim's death. This is in the tradition of the Black press, which connects individual instances of anti-Black violence together to illustrate racism at a larger scale [7, 49]. By repeatedly saying the names of past victims of police violence, the names themselves become signifiers of the same topics, issues, and narratives that are signified by the racial justice movement more generally [6]. Notably though, hashtags and other social media text are implemented through a technical infrastructure that makes data persistent and immediately searchable [50], meaning that online invocations of these names are connected at a scale that was not previously possible. By "saying their names" in each new viral case of police violence then, networked counterpublics are able to more effectively draw upon and expose the cumulative history of anti-Black violence [7–9], lending an individual viral hashtag the narrative weight of all preceding hashtags. While the tactic of connecting instances of anti-Black violence is not new to activists, it is dramatically easier for them to do it through the affordances of social media.

The death of George Floyd in May 2020 caused historic levels of attention to the Black Lives Matter movement and racial injustice [1, 4]. As a pivotal instance of police murdering a Black man that was discussed at unprecedented volumes online, we start by asking:

1. How was discussion of George Floyd, his death, and the following protests amplified and given attention on Twitter?

As discussed, individual instances of police violence, including the murder of George Floyd, do not stand in isolation, particularly online—even at that moment, the deaths of Ahmaud Arbery and Breonna Taylor earlier that year played a notable role in the ensuing conversations. So we further ask:

2. To what extent did people "say their names" and connect George Floyd to past Black victims of police violence?

Finally, since George Floyd draws on the cumulative experience and narrative weight of bearing witness to anti-Black violence, we ask:

3. How have Black victims of police violence been amplified and given attention over the last decade on Twitter, particularly around viral moments?

By answering this set of research questions we both detail a critical specific moment in the history of the racial justice movement, and connect it to a broader history.

## 3 Data and methods

### 3.1 Police-involved deaths of Black victims

Significant attention has been given to the deaths of Michael Brown, Sandra Bland, Philando Castile, Breonna Taylor, George Floyd, and other Black victims of police violence. Unfortunately, their cases are the exception, rather than the norm: most victims of police violence never receive considerable attention on social media, if any. So although it is important to characterize those incidents that have become emblematic of the Black Lives Matter

movement, centering our analysis around *only* them would overlook the many cases of police violence against Black communities that never made their way into mainstream conversations.

Instead, we start from a list of police-involved deaths more broadly and use that to measure the attention that has been given to the names of Black victims. Because of the irregular reporting of police-involved deaths [51, 52], there is no official, complete accounting of victims of police violence, and federal crime data lacks the granularity that is needed to analyze specific incidents of police violence. To address this, we draw from the Fatal Encounters database, a third-party database of people killed during interactions with police officers. It contains records of over 29,000 people killed by police officers since 2000, documented through a mix of web scraping, manual investigation, public records requests, and crowdsourcing by paid researchers and volunteers [53]. It includes both Black and non-Black victims of police violence, and includes the date of the incident that resulted in their death and the cause of death.

To align the Fatal Encounters database with our social media data, we consider deaths occurring from January 1, 2009 onward. We focus specifically on the deaths of Black victims. To direct our analysis towards deaths that are directly caused by police actions, we exclude suicides and vehicular deaths. We also establish several criteria for inclusion based on the names of the victims. First, we remove names that received a measurable amount of attention in the 10 days *prior* to the death of a Black victim with that same name (e.g. Michael Myers, George Bush). We say that a name received *measurable attention* when it was among the top million 2-grams for a day, and provide more detail in the next section. Next, we also manually remove 12 additional names, primarily those shared with famous athletes and two names shared with police officers involved in high-profile deaths related to Black Lives Matter (Darren Wilson and Thomas Lane). For duplicate names in the Fatal Encounters database, we attribute all mentions of a name to the earliest incident that pertains to that name. In S1–S4 Tables, we list all of the names that were excluded from the analysis due to these steps. Finally, we also manually add 15 names significant to the Black Lives Matter movement that were not in the Fatal Encounters database, such as those whose deaths were not caused by direct police action (e.g. Trayvon Martin, Ahmaud Arbery), those whose deaths occurred before the database's timeframe (e.g. Emmett Till), and those who survived serious police violence (e.g. Rodney King). Conducting our analyses without these names would result in an incomplete picture of online attention to anti-Black violence in the United States. We provide the full list of manually included names in S5 Table. Together, these preprocessing steps yield 3,737 records.

## 3.2 Mentions of victims' names on Twitter

Because names are important signifiers in the Black Lives Matter movement [3, 6, 46–48], we measure how much attention has been given to the names of the victims we identified from the Fatal Encounters database since their deaths on Twitter. Twitter has been a particularly important platform for the emergence of Black Lives Matter [3, 6], in part because journalists, pundits, political elites, and others with offline influence frequently look to it for emerging conversations and trends. However, we acknowledge our choice to study Twitter is also one of data access and that it comes with limitations. First, Twitter data reflects a non-uniform and non-representative subsample of opinion. During the time period in which this study is conducted (over a decade), Twitter's user base grew by a factor of nearly 100. The demographic makeup of the population it reflects varies widely during this time, including the proliferation of algorithmically generated accounts and content. Second, changes to how Twitter visually and algorithmically presents content to individuals have led to unobserved and unquantifiable effects in user behavior. These changes include, for example, human-curated trending topics, promoted content, non-chronological feeds, and a variety of platform design adjustments to

increase engagement. In what follows, we quantify collective attention and amplification with the knowledge that the lens through which we do so is complex and in flux.

These caveats in mind, we use the first and last name of each victim in our dataset to create a set of two word phrases, or 2-grams. For each name, we query its frequency and rank over time using the Storywrangler API [54, 55]. Storywrangler uses a 10% sample of English-language tweets to measure how often words and phrases (also known generally as *n*-grams are used on Twitter [56]. For a given *n*-gram, Storywrangler returns a time series of its frequency over time and a time series of its frequency rank over time. The rank is computed for each day by comparing how often a word was used compared to all other *n*-grams of the same length for that day. We retrieve these time series for each victim's name, starting from the date of their injury that resulted in death as recorded in the Fatal Encounters database.

Importantly, Storywrangler only indexes time series data when a particular *n*-gram was among the top million most frequently used *n*-grams of that day. Of the 3,737 names we transcribed from the Fatal Encounters database, 2,603 (69.6%) were never used often enough—neither on the day of their death nor on any day following—to rank as the one millionth most frequently used 2-gram on Twitter or higher. For deaths occurring in 2010, this threshold was roughly 20 mentions on a single day, and in 2020 the threshold was closer to 200 mentions.

Unless stated otherwise, in the following analyses we always characterize the attention given to those who ranked within the top million 2-grams. We say those individuals received *measurable attention*. While this subset is significantly larger than the emblematic names of Black Lives Matter—a fact which allows us to analyze how names are used in the movement in greater detail than we would be able to otherwise—we emphasize that the definitive majority of Black victims of police violence are never mentioned widely on Twitter.

In addition to the time series data for the names, we collect similar data for the phrase "Black Lives Matter" and the hashtag #BlackLivesMatter. We couple that data with measurements made by the Hedonometer, a dictionary-based instrument designed to provide a macro-level approximation of the "happiness" expressed in tweets [57]. To better interpret how the expressed happiness changed in the wake of George Floyd's death, we also use word shift graphs [58] to measure how particular words contributed to fluctuations in the expressed happiness compared to the previous week. See the Supplementary Materials for further detail on the sentiment analysis methods. Our collection and analysis of the data in this study complies with the terms and conditions of the data sources.

### 3.3 Measures of attention and amplification

We use the frequency and rank time series from Storywrangler to define several different measures of collective attention and amplification. We start by noting that raw frequencies of names are subject to fluctuations in how many tweets were written in general on Twitter, which can vary widely in both short- and long-term time windows. Instead, for each day we calculate each name's relative frequency, which is the name's frequency normalized by the total frequency of all words on Twitter for a particular day—where we note that for all of the measures described below, we always compare to *n*-grams of the same length. Equations reflecting the definition of relative frequency $p_{\tau,t}$ can be found in the Supplement.

We use the relative frequencies to construct several other measures of attention. First, we define the proportion of days that a name received attention as the proportion of days since the date of death that a name was within the top million 2-grams. Second, we say that the *peak attention* given to a name is the highest relative frequency with which it was used, i.e. the maximum of $p_{\tau,t}$ across all days *t*. We use the peak attention to construct the *normalized attention*, a value between 0 and 1, where 0 indicates a name was not used within the top million 2-grams

for a day, and 1 indicates the day that the most relative attention was given to a name since the date of death (see Supplement for formula).

Similarly, we define two further measures of attention by measuring the rank of how frequently each name was used on each day $r_{\tau,t}$, and its peak rank, the minimum of $r_{\tau,t}$ across all days $t$ (since lower values of $r_{\tau,t}$ indicate a higher rank). The rank and frequency metrics give related, but different measures of collective attention. The frequency-based measures of attention give a sense of how much a name was used relative to all the other times it was used on Twitter. The rank-based measures give us way of interpreting how high or low that attention was compared to all other 2-grams.

We measure the amplification of each name by distinguishing between how often they were used in originally authored tweets (OT) and retweets (RT), where we include the novel part of quote retweets among originally authored tweets. We operationalize amplification as the ratio between the two frequencies for a given name, namely $R_{\tau,t}$. This ratio is 1 when a name is used equally often in originally authored tweets and retweets. If $R_{\tau,t}$ is greater than 1, then the name is amplified via retweets more often than it is written itself, and vice versa if $R_{\tau,t}$ is less than 1.

Note that this operationalization of amplification differs from one of the most standard approaches to measuring amplification, which is simply counting retweets. By comparing to how much a name is being written in originally authored tweets, we establish a baseline that allows us to discern to what extent the amplification $R_{\tau,t}$ is more or less than what we would expect given how much that name is being spoken about on Twitter. When a name does not receive measurable attention—that is, when it does not rank within the top million 2-grams—we say that $R_{\tau,t}$ is 1, since it was not measurably retweeted more nor less than the inattention it received.

It is important to note that retweets have become increasingly common over time in English language tweets [60]. This means that if we look at $R_{\tau,t}$ over longer time windows, as we do here, names that appear later in our study frame will appear to have been more likely to be retweeted. However, the long-term increase in the rate of retweets is confounded by other factors such as changes to Twitter's design and algorithmic curation. To account for this, we define the *relative social amplification* $R_{\tau,t}^{\mathrm{rel}}$ [55, 60] (see Supplement for formula). We normalize the amplification of a name by the ratio of how much all English language is used in retweets versus originally authored tweets on any given day. So, unlike $R_{\tau,t}$, the relative social amplification $R_{\tau,t}^{\mathrm{rel}}$ is comparable over wide time frames on Twitter, allowing us to compare the amplification of names to themselves and one another over time.

## 4 Results

### 4.1 Unprecedented attention in the wake of George Floyd's death

We start by characterizing the attention given to and amplification of George Floyd's death in late May 2020, reaffirming the ways in which it was a pivotal moment in the history of both Black Lives Matter and Twitter more broadly. George Floyd's death on May 25th, 2020 was followed by a massive increase in tweet volume lasting until about June 7, with a peak on June 2nd (see middle panel of Fig 1). Approximating from the 10% Decahose random sample of tweets, we estimate that an average of 197 million tweets were sent per day during that period, with up to about 219 million tweets per day between May 31 and June 2, 2020. Those latter three days constitute the 2nd, 3rd, and 4th days with the most tweets overall compared to all other days in our Decahose sample, dating back to 2009. Further, they are the top three days in Twitter history in terms of the number of retweets that were sent. For comparison, we estimate about 157 million tweets were authored per day from May 1st to May 26th, 2020.

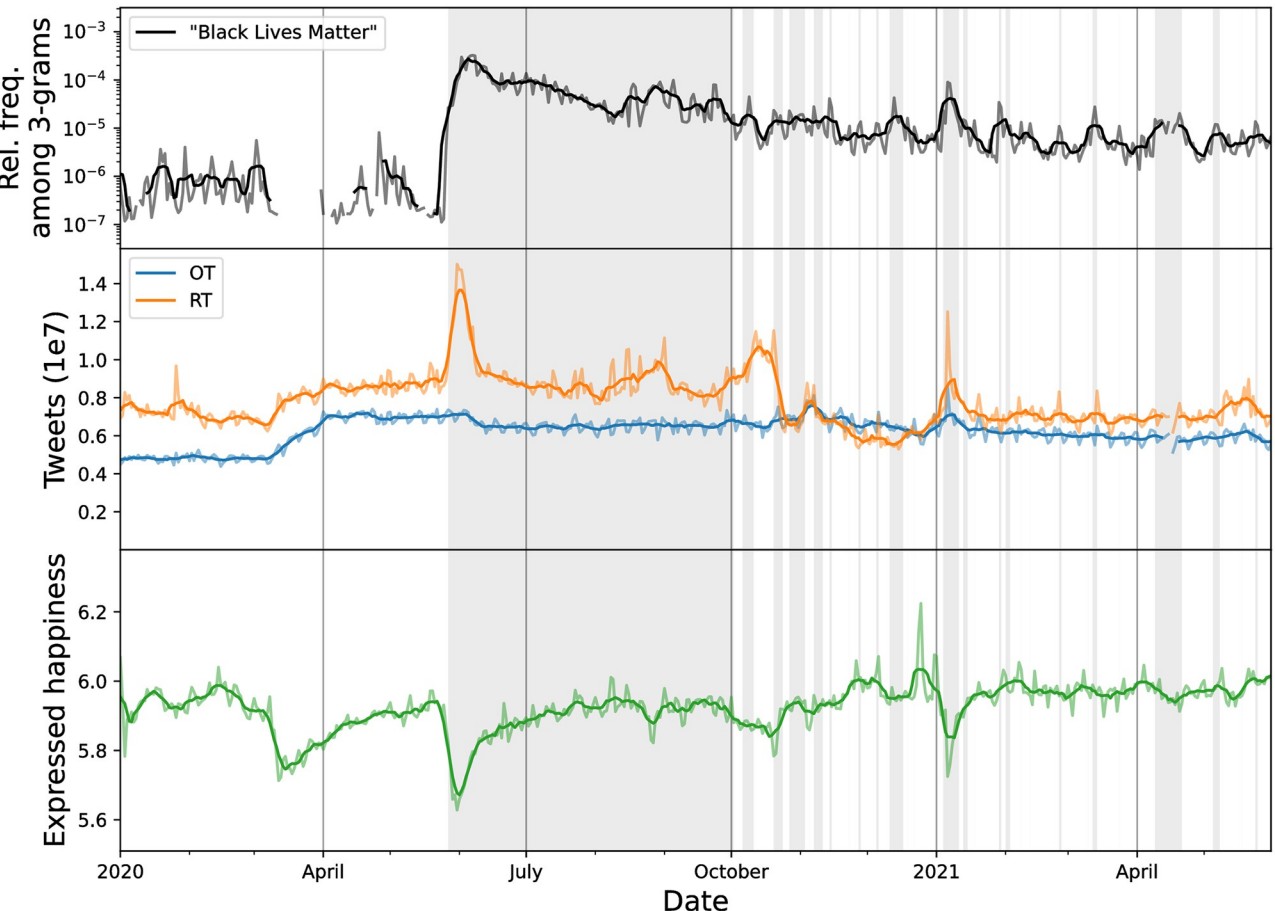

**Fig 1. English tweet volume and expressed happiness (random 10% sample), 2020–2021. Top)** Relative frequency of the phrase "Black Lives Matter". The time series is absent at times where "Black Lives Matter" did not receive measurable attention, i.e. it was not within the top million 3-grams. The gray shaded areas represent periods of time when "Black Lives Matter" was among the top 5,000 most used 3-grams per day. **Middle)** Daily counts for originally authored tweets (blue, OT) and retweets (orange, RT) reveal a spike period in late May and early June 2020 during which retweet activity set all time records. The sudden decrease in retweets in October 2020 is attributable to a platform-wide design change to retweets on Twitter [59]. **Bottom)** Average expressed happiness of English-language Twitter shown by day. Drops in expressed happiness corresponding to the pandemic (March 2020) and the Capitol insurrection (January 2021) are both apparent. Less happiness was expressed in the period following George Floyd's death than any other point in Twitter's history. For all panels, light lines indicate the raw frequency and sentiment values, and darker bold lines indicate 7-day rolling averages.

As shown in Fig 1 and detailed further in Table 1, the increase in tweet volume was driven almost entirely by retweets. The average number of originally authored tweets per day increased from 69.7 million between May 1st and May 26th to 71.2 million between May 27th and June 7th—an increase of only 2.1%. Meanwhile, the average number of retweets per day

**Table 1. Change in average daily tweet volume in May 2020 and the subsequent spike period by tweet type.** Messages are separated into originally authored tweets, retweets, and their sum (in millions) between May 1–26 and May 27–June 7, 2020.

| Tweets (millions) | May 1–26 | May 27–June 7 | % Change |
|---|---|---|---|
| Original | 69.7 | 71.2 | +2.1 |
| Retweets | 87.5 | 126 | +44 |
| Total | 157 | 197 | +25 |

rose from 87.5 million between May 1st and May 26th to 126 million between May 27th and June 7th, yielding a dramatic 44% increase. This spike period is characterized by a simultaneous spike in the discussion of Black Lives Matter (top panel of Fig 1).

The increase in tweet volume during the spike period resulted in "George Floyd" being the 7th most used 2-gram on Twitter on May 29th, 2020. To help convey just how exceptional this rank is [61], we report that only the functional 2-grams "!!", "of the", "??", "in the", ", and", and "to be" appeared more frequently than "George Floyd", with "to the", "on the", and ". I" rounding out the top ten. Only a few dozen other 2-gram proper names have reached similar levels of attention over approximately five thousand days of Twitter between 2008 and 2022. Others include "Muhammad Ali" who reached a rank of 2 on June 4th, 2016, the day after his death, and "Donald Trump" who reached a rank of 6 on November 9th, 2016 the day following his election as president of the United States. Among those who have reached such stratospheric levels of attention before May 2020, George Floyd is the only one who was unknown to the public earlier the same week.

To give further grounding, in separate analyses of 1-gram distributions [61], function words durably populate the top 1000 1-grams in English. Over the period of 2015 to 2020, only the K-pop band BTS and Donald Trump maintained ranks in the top 300 on a near daily basis. An analysis of country names used on Twitter in the year 2018 showed that 'America' was the most common with a rank of around 1000. In sum, reaching the top 10 1-gram and 1-gram for a day on Twitter is rare, and sudden jumps to such heights signify a world-scale event.

The increased volume in tweets corresponded with a decreased expression of happiness on Twitter (see bottom panel of Fig 1). Based on our 12-year Decahose sample, May 31st, 2020, six days after George Floyd's death, was the day with the least happiness expressed in Twitter's history. The average happiness expressed on that day was 5.628, while the average happiness expressed in the preceding seven-day period was 5.803. For reference, the average happiness expressed from 2009 to 2021 is 6.012 with a standard deviation of 0.062. This drop in happiness was unique not only in its intensity, but also its duration. Prior to the COVID-19 pandemic, nearly all decreases in happiness caused by tragedies—like other police violence events, mass shootings, and celebrity deaths—have lasted only about a single day before returning to "normal" levels [62, 63]. It is difficult to gauge a "normal" level of happiness expressed on Twitter, but the happiness expressed in June 2020 did not reach a level of 5.9 or greater again until June 21st (Father's Day). The only other period with such an evidently long recovery was during March and April 2020 when the SARS-CoV-2 coronavirus pandemic dramatically altered daily life in the United States.

During the slow regression back toward "normal" happiness on Twitter, discussion shifted from the event of George Floyd's death to the protests following it. Examining the individual words that contributed most to the decrease in expressed happiness, we find "murder," "killed," and "death" had strong contributions shortly after his death. Words such as "violence," "protest," and "terrorist" then began to show stronger contributions in the following days, suggesting that the collective conversation began to move away from Floyd's death and toward the protests themselves. S1–S4 Figs fully summarize the words that contributed the most to the decreased expression of happiness.

Complementing these observational findings online, survey-based measures of mood in the United States during this period reveal unprecedented emotional lows offline as well. [64] analyzed about half a million Gallup polls and U.S. Census responses, concluding that "in the week following Floyd's death, anger and sadness increased to unprecedented levels in the US population. During this period, more than a third of the US population reported these emotions." Black respondents were more likely to report symptoms of anxiety, and the authors

estimate nearly one million Black Americans would have screened positive for depression during this period.

## 4.2 Resurgence of past names

In the wake of George Floyd's death, the historic amount of attention was not just given to him nor even just to the subsequent protests. It also extended to past Black victims of fatal police violence. To show this, we measure the extent to which their names received heightened attention during and shortly after the spike in Twitter activity from May 25th to June 7th. For each name, we calculate the average relative frequency with which it was used thirty days before the spike. We then compare that to its average relative frequency during the spike, and thirty days following the spike. For reference, we compare the average change in relative frequency during the George Floyd spike to the average change during three other important spikes in Black Lives Matter history: the emergence of #BlackLivesMatter around the deaths of Michael Brown, Tamir Rice, and Eric Garner (November 24th–December 8th, 2014), the introduction of #SayHerName and the death of Sandra Bland (July 13th–July 26th, 2015), and the deaths of Philando Castile and Alton Sterling (July 5th–July 13th, 2016). We also compare to the "Unite the Right" rally in Charlottesville, Virginia (August 12th–August 22nd, 2017), during which time use of the hashtag #BlackLivesMatter spiked but not due to a police-involved death.

More attention was given to Black victims of fatal police violence during the spike following George Floyd's death than any of the other spikes, and that attention persisted longer following the spike (see Table 2). The average change in average relative frequency is an order of magnitude larger during the George Floyd spike than the ones following the deaths of Sandra Bland, Philando Castile, and Alton Sterling, and 34% larger than when #BlackLivesMatter first saw widespread use in late 2014. Relative to the pre-spike periods, while the average change in attention to names decayed by an order of magnitude after the initial spike of #BlackLivesMatter and by 59% following the deaths of Philando Castile and Alton Sterling, it only declined by 33% as the spike following George Floyd's death subsided. As expected, use of Black victims' names did not increase significantly during the "Unite the Right" rally, as police-involved deaths were not the topic of concern. We find these results are robust even if we vary the pre- and post-spike time windows to be anywhere from 7 to 90 days long (see S6 Table).

Further, the increase in attention to victims of fatal police violence was not just caused by attention to names that are most emblematic of the Black Lives Matter movement (e.g. Michael

**Table 2. Resurgent attention to past victims of police violence following George Floyd's death and other periods of interest to Black Lives Matter.** We consider four periods of spikes in attention relevant to #BlackLivesMatter: November 24th–December 8th, 2014 (Deaths and non-indictments in the cases of Michael Brown, Tamir Rice, and Eric Garner), July 13th–July 26th, 2015 (death of Sandra Bland), July 5th–July 13th, 2016 (deaths of Philando Castile and Alton Sterling), August 12th–August 22nd, 2017 ("Unite the Right" Charlottesville rally), and May 25th–June 6th, 2020 (death of George Floyd). The number of names that received increased attention during a spike period is reported, as well as the percentage of those that had not received any measurable attention in the 30 days prior to the spike. The average change in average relative frequency is calculated for the difference between 30 days before the spike period and during it, and 30 days before and after it. Statistical significance is indicated by * for $\alpha = 0.05$ and ** for $\alpha = 0.01$.

| Spike Period | # Names with Increased Attention | % Names w/No Atten. 30 Days Before | Avg. Diff. in Rel. Freq. Spike—Before | Avg. Diff. in Rel. Freq. After—Before |
|---|---|---|---|---|
| Nov. 24–Dec. 8, 2014 | 81 | 71.6% | 3.82e-08 | 3.37e-09 |
| Jul. 13–Jul. 26, 2015 | 36 | 55.5% | 5.48e-10 | 4.49e-10 |
| Jul. 5–Jul. 13, 2016 | 68 | 66.1% | 6.83e-09 | **2.76e-09**\* |
| Aug. 12–Aug. 22, 2017 | 34 | 32.3% | -1.40e-10 | -7.88e-10 |
| May 25–Jun. 6, 2020 | 186 | 72.3% | **5.14e-08**\*\* | 3.43e-08 |

Brown, Sandra Bland). The names of 186 victims were used more during the spike following George Floyd's death than they were in the thirty days before. This is more than twice as many victims that were mentioned in any of the other spike periods that we look at here, and over 70% of those names had not been mentioned at all in the month proceeding Floyd's death. In S7 and S8 Tables, we display the names of all those who received more attention on average during the spike than prior to it.

This is not to say that prior surges in attention to Black Lives Matter have not been effective at highlighting instances of police violence, nor that they did not have any lasting impact. Rather, the increased attention following George Floyd's death should be understood as the culmination of over 6 years of movement building since #BlackLivesMatter first gained traction following the death of Michael Brown in 2014. Of note, despite some notable exceptions, the most invoked names before and after George Floyd's death were names of men, underscoring the #SayHerName call to action and its ongoing importance.

## 4.3 Long-term trends of attention and amplification

To put May and June 2020's exceptional levels of attention to police violence in context and understand how that moment built off the foundations laid by the Black Lives Matter movement and other activists, we step back and view how attention has been given to police-involved deaths over the past 12 years on Twitter. We take this view from two different perspectives: that of attention and that of amplification. Together, these paint a detailed picture of how individuals on Twitter have reiterated the names of Black victims of police violence.

**4.3.1 Collective attention.** In Fig 2, we show how emblematic names of the Black Lives Matter movement have been used on Twitter across the past decade. Although #BlackLivesMatter did not gain traction until 2014, it was created in 2012 following the death of Trayvon Martin. Even without the hashtag, though, we see that his name has been consistently used since his death. With the deaths of Eric Garner, Michael Brown, Tamir Rice, Freddie Gray, and Sandra Bland, we see increases in attention to different prominent victims of fatal police violence, and that the attention to those names was sustained as the movement gained its ground. While there were a number of high-profile incidents over the latter half of the decade, there is a stark vertical band around May and June 2020 following George Floyd and Breonna Taylor's deaths in which many of the most recognizable victims of police violence all received high levels of attention. This is one visual indicator that reaffirms the resurgence in names in the wake of George Floyd's death.

However, as we have discussed, the most high-profile incidents of fatal police violence are just a fraction of all incidents. In Fig 3, we show the spectrum of how much attention has been given to all Black victims of police violence over time. Again, we see a notable increase in attention around late 2014, as the hashtag #BlackLivesMatter gained traction. Compared to pre-2014, the amount of attention given to names has been consistently higher, as indicated by the increase in dark cells. The strong band of color around May and June 2020 is again visible, showing that not only did the most prominent names gain increased attention (note the band on the right-hand side of the plot), but also many other names did as well, at various frequencies. Note, there is a dark band on the left-hand side of the plot as well: while the names of many victims have been mentioned on Twitter and increasingly so over time, there are also many others do not see such widespread attention.

In Fig 4, we unpack the relationship between the number of days that different names have received attention, and how widely they have been discussed, as measured by their peak rank compared to all other 2-grams on Twitter for a given day. Among those with the most and widest attention, we see the emblematic Black Lives Matter names, like Breonna Taylor, Walter

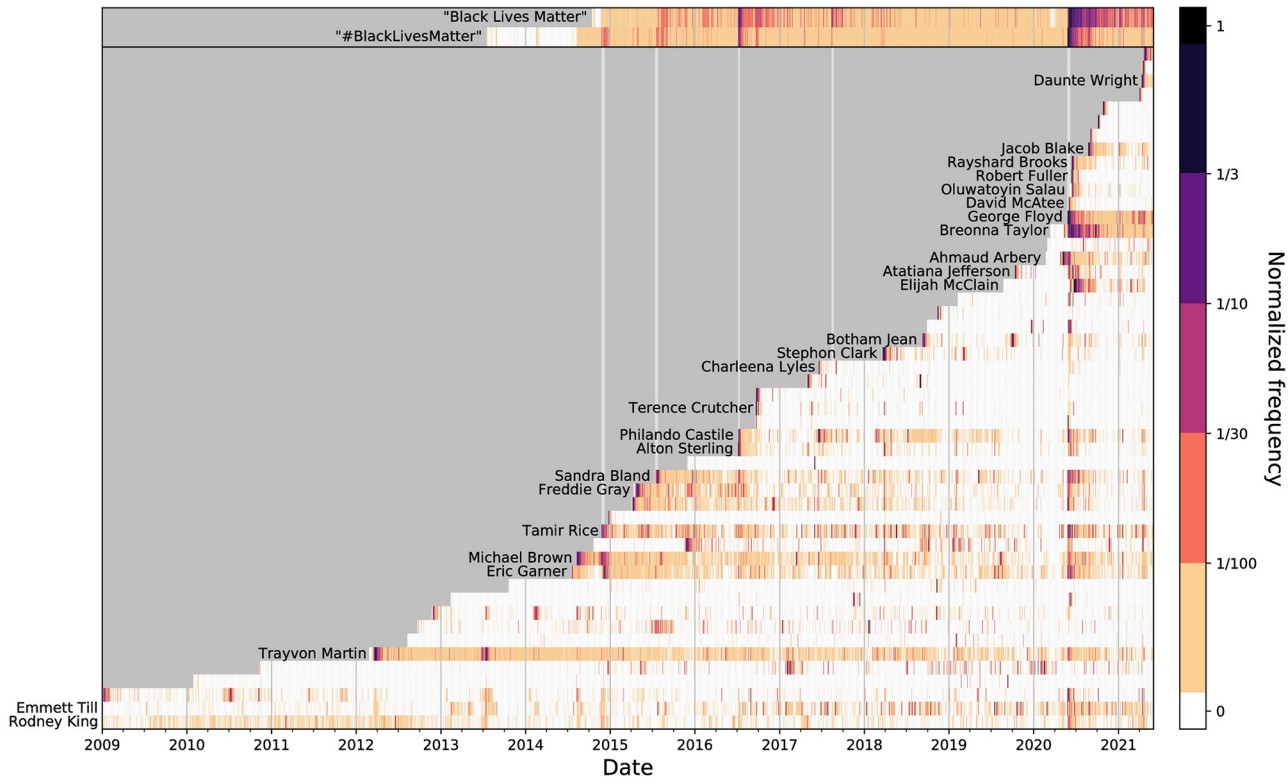

**Fig 2. Attention towards emblematic names of Black Lives Matter. Top)** Heatmap time series of the normalized relative frequency of "Black Lives Matter" and "#BlackLivesMatter". **Main)** Heatmap time series displaying the normalized attention $\widehat{p_{\tau,t}}$ over time for the top 50 names in the combined database by peak rank. The top 25 names are labeled. Darker colors indicate more attention, where the normalized attention is 1 (black) on the date of peak attention. Dark bands around June 2020 are observed across many of the rows, indicating a resurgence in attention for many of the emblematic names.

Scott, Alton Sterling, and Sandra Bland. The plot also reveals the number of names that do not receive those heights of attention. Most police-involved deaths of Black victims generally have not consistently received attention, nor have they they had high peak ranks. Given that most incidents receive their peak attention within a week of their occurrence (see S5–S7 Figs, this makes it difficult for them to ever receive high attention if they do not so immediately. This is particularly clear among the incidents that occurred around 2010, which we see generally have lower peak ranks, and lower proportions of days with attention. There are notable exceptions in either direction of the proportion of days with attention and the peak rank. For example, some (e.g. Darius Smith) received a relatively high amount of attention (i.e. high peak rank) at one point, but have not been given attention for many days in general. Others (e.g. Patrick Warren, Sincere Pierce) never reached a relatively high level of discussion, but have still been more consistently discussed since their deaths than others. In S8 and S9 Figs, we further detail the relationship between proportion of days with attention and peak rank by gender. Overall, the figures demonstrate the range of attention that has been given to Black victims of police violence, and how that attention extends beyond the most emblematic names of the Black Lives Matter movement.

**4.3.2 Relative social amplification.** As with attention, amplification of the names of Black victims of fatal police violence has also varied over the past decade on Twitter. In Fig 5, we show the different trajectories of how some of the most emblematic names of the Black Lives Matter movement have been amplified over time. For each name, we measure both the

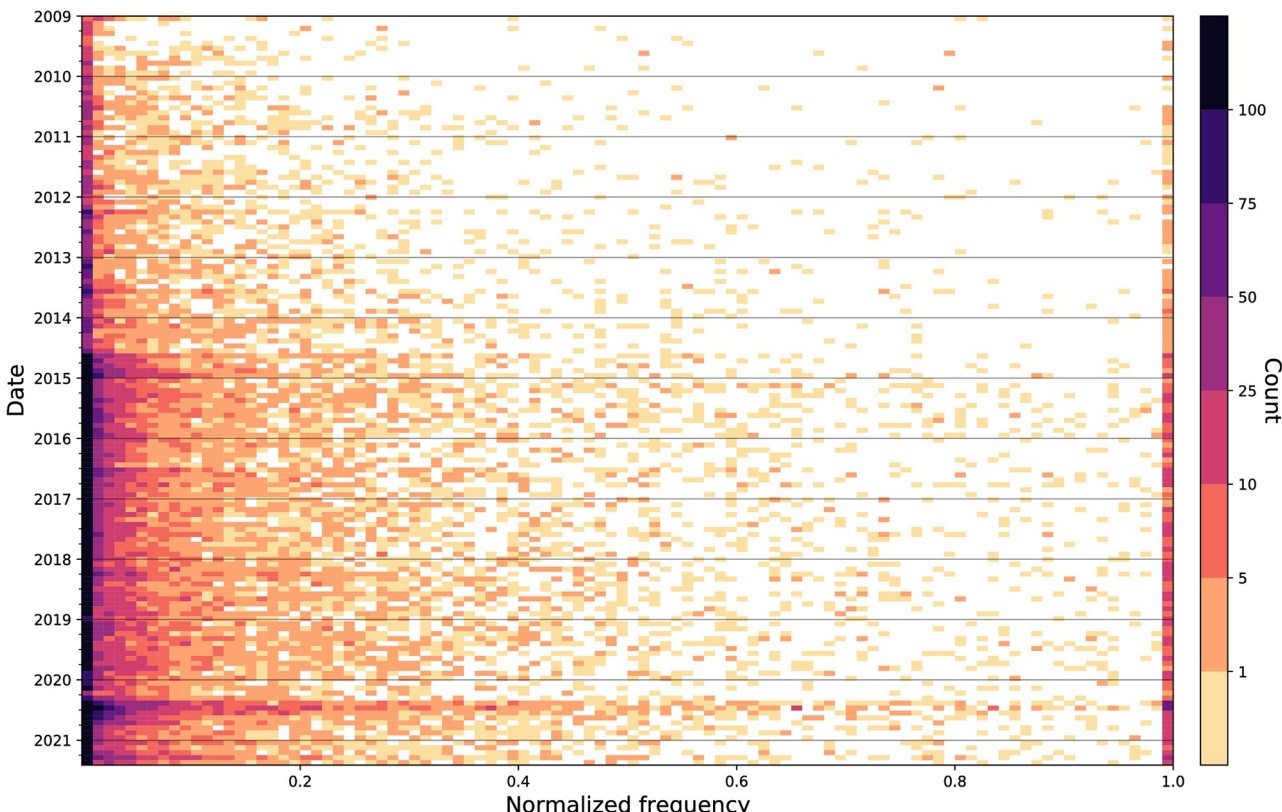

**Fig 3. Collective attention toward names over time.** Histograms over time showing the amount of normalized attention $\widehat{p_{\tau,t}}$ given to Black victims of fatal police violence from January 1, 2009 to May 31, 2021 (intervals of about 30 days). Each cell indicates the number of names that received that particular amount of attention in that moment of time. There is a band around May–June 2020 across many different frequencies, indicating a variety of attention for many names during that period.

proportion of tweets containing the name that were originally authored tweets and retweets, and how amplification of that name changed relative to English language as a whole. Like attention, we can see that the amplification of Trayvon Martin clearly predates the widespread use of #BlackLivesMatter, and his name has been consistently amplified since his death. A feature of some of the time series—including his, but also others such as Michael Brown, Eric Garner, and George Floyd—are two distinct peaks in attention with regard to their rank $r_{\tau,t}$. The dual peaks for these cases correspond to the initial attention given to the victim's death, and the later (non-)indictment or trial. For some, the peak rank of their name on Twitter is higher at the time of legal judgement—#BlackLivesMatter, for example, only gained traction after the non-indictment of Darren Wilson for the death of Michael Brown, and Michael Brown's name was used more then than at the time of his death. Further examples of these amplification trajectories are shown in S10 Fig.

As we see by these examples, amplification ebbs and flows, even for the most emblematic names. Again, given that most police-involved deaths are not those that go viral through #BlackLivesMatter, we look in Fig 6 at the amplification of the names of those who received measurable attention. In late 2014, we see more amplification of a wider variety of names compared to the pre-#BlackLivesMatter era. We see this amplification consolidate in two ways. First, over time, more names receive more relative social amplification. In particular, since 2014, most names of Black police violence victims that receive measurable attention on Twitter

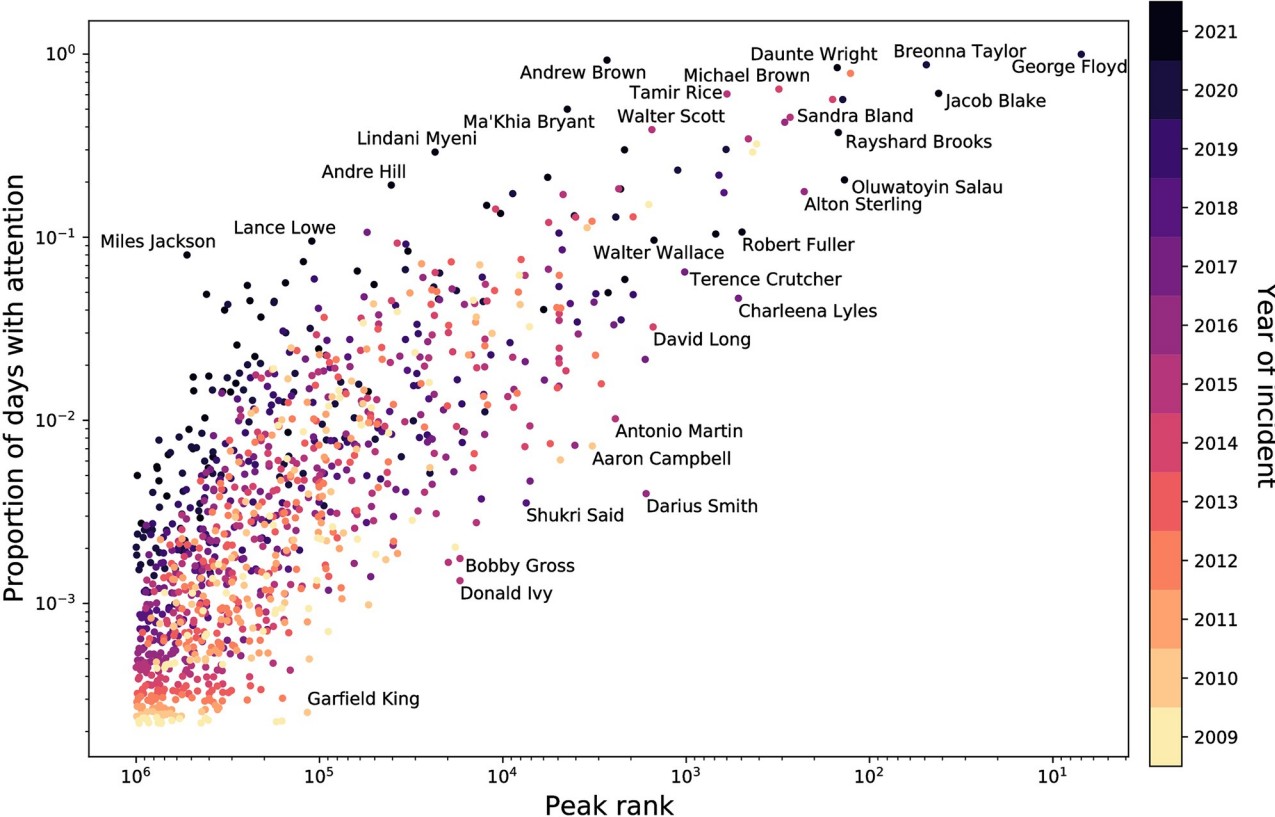

**Fig 4. Proportion of days with attention versus the peak rank of names.** Each point is colored by the year of the event or death. There is a moderately strong trend—as peak rank increases, the proportion of days mentioned increases. Names with a relatively high proportion of days with attention and a relatively high peak rank tend to be more recent deaths.

have a relative social amplification $R^{\text{rel}}_{\tau,t}$ greater than 1, meaning that they are amplified (retweeted) more than other English 2-grams. In other words, the names of Black victims of fatal police violence receive more amplification than we would expect compared to English language amplification generally. We also see a consolidation of amplification: names generally are no longer amplified more than about 1.5 times more than other English $n$-grams, as compared to the early 2010s, during which time the relative social amplification could be upwards of three times higher (or more, see S11 and S12 Figs. This is likely a result of the increase in the proportion of tweets that are retweets in English in general [60]. Although the relative social amplification may not reach as drastic of heights, whether that be due to platform design changes or alterations to how Twitter curates timelines, there is a much higher number of names that receive amplification, and—if a name receives measurable attention on Twitter— most of the time that amplification exceeds what we would expect otherwise.

Finally, we compare the average relative social amplification of each name $R^{\text{rel}}_{\tau,t}$ to its peak rank and the proportion of days it received measurable attention (see Fig 7). On average, the relative social amplification of most names is 1, meaning that they are amplified as commonly as we would expect relative to how much all other English language is amplified. We see though that names with the highest proportions of days with attention and highest peak ranks are those that receive the most amplification on average. This makes the barrier of entry into the mainstream public sphere [31] clear: it is difficult for a name reach a high level of visibility

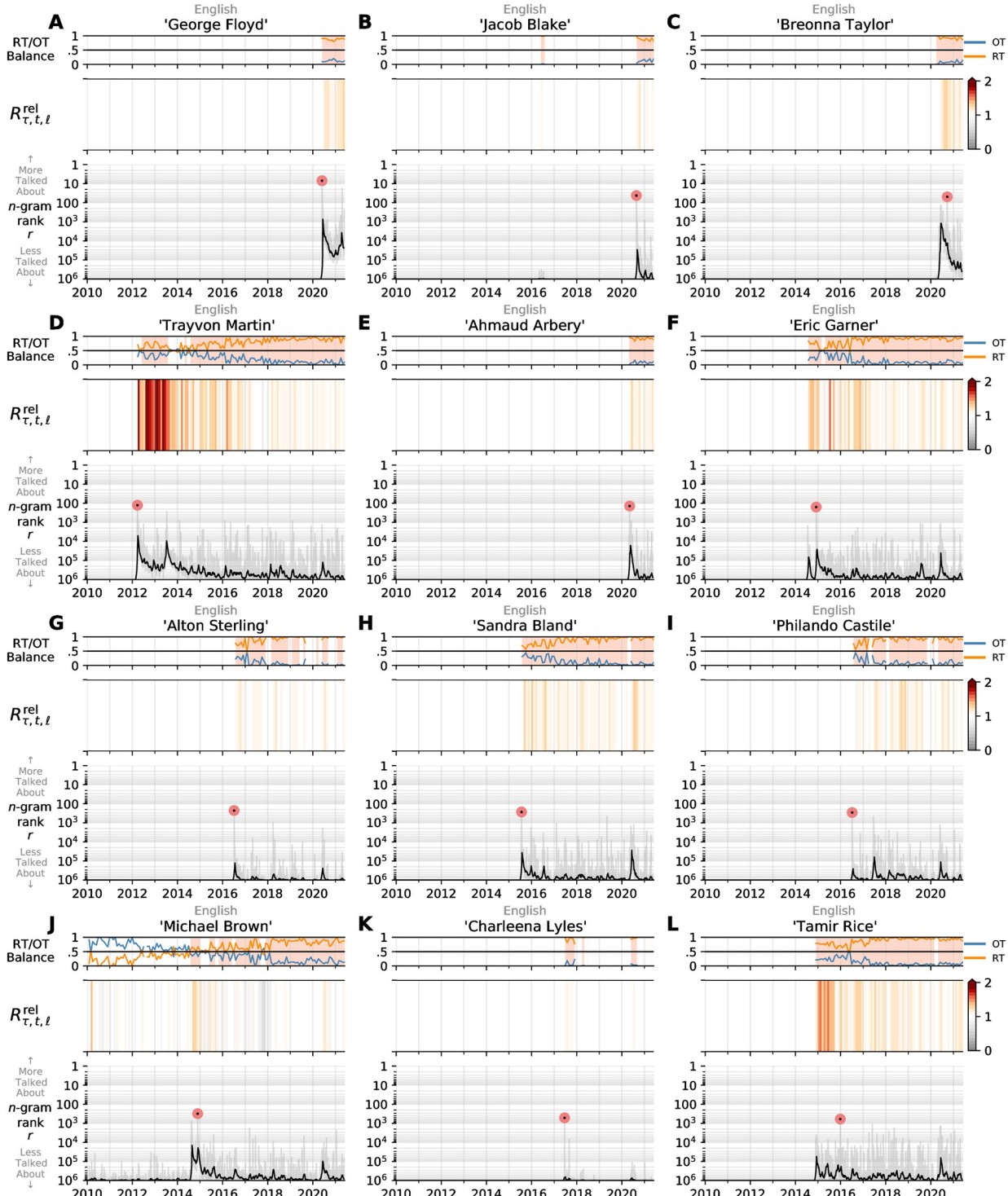

**Fig 5. Amplification of emblematic names of Black Lives Matter.** Each panel visualizes amplification and attention in three ways: **Top)** Time series of the proportion of mentions of a name that were from originally authored tweets (blue) and retweets (orange). **Middle)** Heatmap time series of the relative social amplification $R^{rel}_{\tau,t}$ of a name relative to all English language. When the relative social amplification is greater than 1, that period is highlighted in orange in the top panel as well; this means that the name is amplified (retweeted) relatively more than other English $n$-grams of the same length. **Bottom)** Time series of a name's daily rank compared to other $n$-grams of the same length. The 7-day rolling average is shown in bold, and the day of peak rank is indicated with a pink dot. See S10 Fig for other selected names.

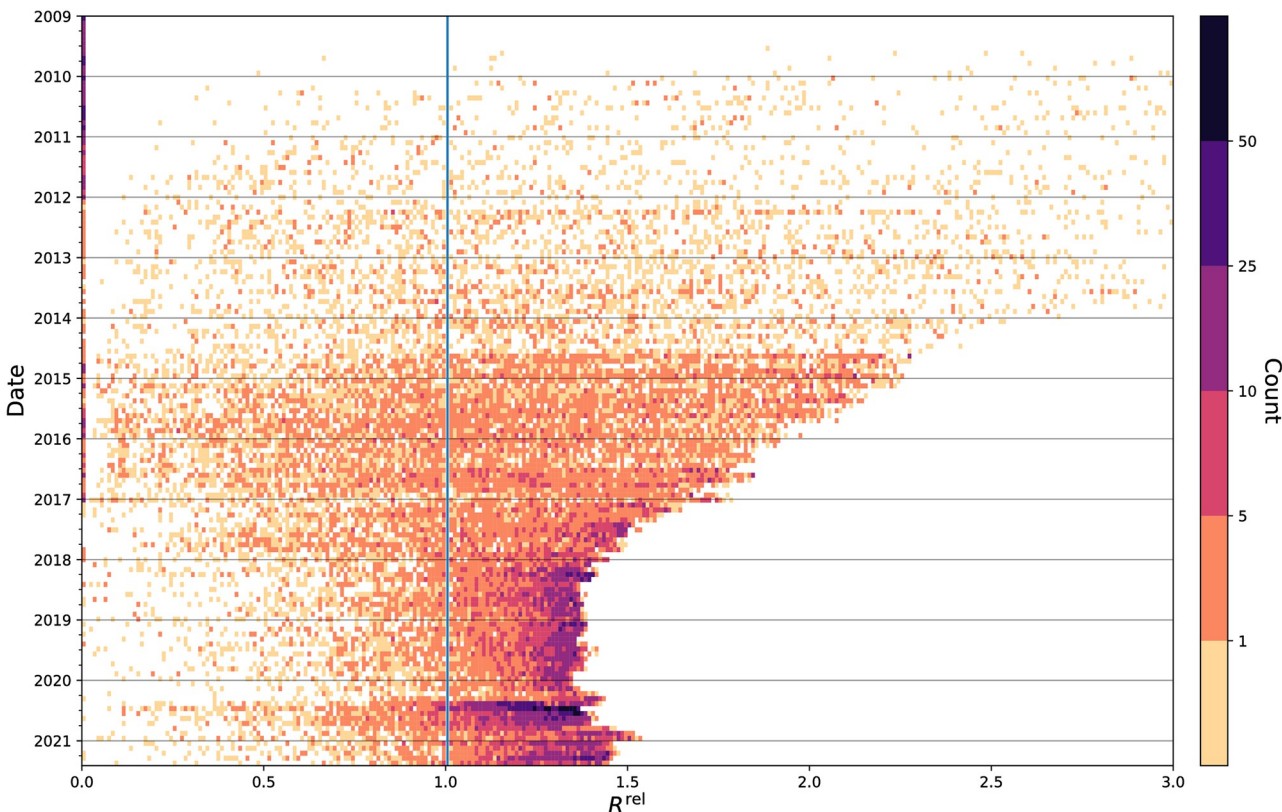

**Fig 6. Relative social amplification of names over time.** Histograms over time showing the relative social amplification $R^{\mathrm{rel}}_{\tau,t}$ given to Black victims of fatal police violence from January 1, 2009 to May 31, 2021 (intervals of about 30 days). Each cell indicates the number of names that received that particular level of relative social amplification in that moment of time. Names increasingly have a relative social amplification higher than 1, meaning they are amplified (retweeted) relatively more often than other English 2-grams. Around May and June 2020, more names take on a high $R^{\mathrm{rel}}_{\tau,t}$ of about 1.4, indicating high social amplification for many names during that period. The maximum relative social amplification for names decreases over time because the proportion of all English tweets that are retweets increases over time [60].

(high peak rank) or consistently receive attention without a consistently high level of relative social amplification. That is, this demonstrates a link between attention and amplification, where it is likely the case that it is both difficult to receive widespread attention without high levels of amplification, and difficult to receive amplification without high levels attention. This likely contributes to why most Black victims of police violence never receive measurable attention on Twitter.

## 5 Discussion

This study contributes an event-based view of how George Floyd was given attention and amplified on Twitter, and a retrospective view of how Black Lives Matter has engaged in the longstanding tactic of invoking Black victims' names to connect new instances of police violence with past ones. In particular, we have shown that there was an exceptional resurgence in attention to past victims following the death of George Floyd. His death had an unprecedented impact on Twitter: it prompted the 2nd, 3rd, and 4th most tweets to be sent per day in Twitter's history, where those three days have the most retweets of any previous day on Twitter. The historic levels of tweeting were matched with a historic dip in happiness expressed on the platform: over 12 years of Twitter data, happiness has never dropped as low as it did on May

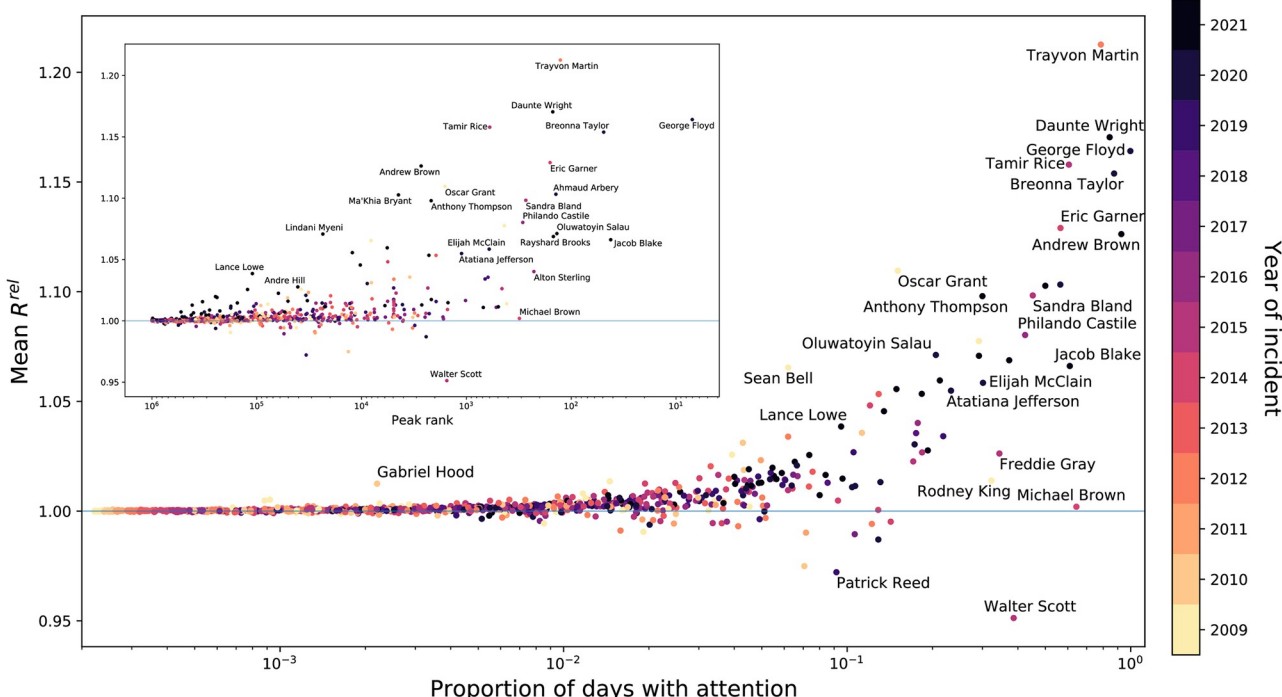

**Fig 7. Mean relative social amplification versus proportion of days with measurable attention and peak rank.** The mean relative social amplification $R_{\tau,t}^{\mathrm{rel}}$ of most names is close to 1, but as the peak rank and proportion of days with measurable attention increase, the range of relative social amplification values widens. Each point is colored by the year of the event or death.

31st, six days after Floyd's death and in the midst of the subsequent protests. The attention to his death though came with exceptional levels of attention to other past victims as well, particularly as compared to prior spikes in attention to the Black Lives Matter movement. Understanding that those levels of attention built off prior surges in attention to #BlackLivesMatter and an even longer history of racial justice activism, we put the surge of attention following Floyd's death within the context of how the names of Black victims of police violence have been given attention over the past 12 years on Twitter.

In light of deeply ingrained systemic racism and police brutality, the Black press, the Black public sphere, and the Black Lives Matter movement memorialize victims by saying their names. Victims' names are used to honor and draw attention to specific individuals and, especially when named in combination, to connect individual deaths with the history of state violence against Black Americans. In this way, names become signifiers of broader injustices, with particular combinations evoking various dimensions of injustice to be addressed. For example, pairing Emmett Till and Trayvon Martin might signify state violence against Black youth, while pairing Sandra Bland and Breonna Taylor might signify state violence against Black women (and its corresponding erasure in media coverage). The evocative nature of names as signifiers allows Black Lives Matter activists and allies to tell complex, intersectional stories about police violence in the limited space of a tweet. Compared to activism prior to the internet then, hashtag activism and the reiteration of names online allows anyone to easily and literally connect different cases of police violence—no printing press, television production, nor organizational infrastructure is needed for others to bear witness. As has always been the the goal of racial justice campaigns, social media helps activists and allies contextualize individual victims within a broader, ongoing history of police violence. Our work demonstrates this

in several ways: we show how the attention given to George Floyd was a rising tide that brought attention to over 185 past instances of police violence, how attention and amplification of Black victims has gradually increased over the last decade on Twitter, and how attention is given to the many more victims that go beyond just the most emblematic names of Black Lives Matter.

The Black Lives Matter movement emphasizes patterns of police violence across events, but research and reporting often still treat it episodically, focusing only on particular viral moments and hashtags. By engaging in a longitudinal retrospective study, our work is an exemplar of how to treat Black Lives Matter as a movement—rather than a moment—through computational research. By looking at the totality of Black victims of police violence, rather than any single one, we can learn more about how signifiers are used in racial justice activism. The extent to which that activism is embedded in a longer history of bearing witness to anti-Black violence raises deeper questions about how other instances of hashtag activism are and are not able to be as impactful as #BlackLivesMatter. The use of hashtags as a rhetorical protest technique is not unique to the Black Lives Matter movement, nor is it the only movement to successfully use them to achieve visibility [6]. However, #BlackLivesMatter is much more exceptional in its ability to achieve sustainability. While the persistence of #BlackLivesMatter certainly depends upon a myriad of factors, including formal social movement organizing, our study suggests that it is also particularly rooted in Black Twitter as a cultural extension of the Black public sphere [23, 26, 27]. Taking this historically contextual view—where the success of hashtag activism relies upon its ability to draw upon deep, long-running movements and narratives—challenges the view of hashtag activism as spontaneously "viral." For example, rather than viewing the similarly exceptional impact of the hashtag #MeToo as a singular one-off success, it should be seen as the culmination of several preceding hashtag campaigns challenging sexual violence—including #YesAllWomen and #TheEmptyChair [6]—that was best able to tap into the centuries-long momentum of feminist movements. While signifying names as hashtags in the tradition of the Black press is particularly unique to #BlackLivesMatter, our work points toward the potential for future work on hashtag activism to connect seemingly instantaneous online discussions of marginalization and oppression to their deeper histories.

While we have focused here on comprehensively characterizing *who* receives attention and amplification, there is more to be done to understand *why* they do: factors like a victim's gender, age, or whether there is video footage of an event may determine the amount of attention that a name receives. Further, it will be important for future work to move from measuring aggregate collective attention to measuring how individual users and groups of users give attention to racial justice. How we understand the attention given to a particular victim may differ depending on whether that attention results from a small number of viral tweets or many original tweets, or whether it comes from liberal or conservative accounts. Of course, there are also questions of how the online attention relates to offline impact. Data on offline racial justice protests and tweet volume around particular victims could be used to quantify how social media is used to organize protests, donations, and petitions for Black Lives Matter and other social movements [65]. Because we started with an extensive list of victims of police violence, rather than focusing just on cases which received national levels of attention, this would contribute to our understanding of how online discussion can catalyze further offline activism without selecting on the most visible instances of online activism. Such attention, online and offline, may be affected by events other than death: birthdays, anniversaries, and trials may also uniquely contribute to resurgences in attention to those who have been killed by police violence. More broadly, the methods used here to analyze attention towards police violence victims could be applied to study other political figures, historical events, and social justice movements.

Our study rests on the Fatal Encounters database, a third-party repository for recording police-involved deaths. This is because there is a glaring lack of government-maintained police violence databases, and there is a clear need for better tracking and reporting of how police use force in the field. Merging the database with the social media data required us to make choices about inclusion and exclusion. Specifically, some of these choices involved attributing mentions of a duplicate name to just the earliest instance of that name, and excluding those who had names that received attention prior to their death, some of which were shared with celebrity figures. More sophisticated methods for determining the context of how a name is being used in a tweet would allow us to more inclusively study victims of police violence.

We reiterate that Twitter is not representative of the general population [66–69], though it is worth emphasizing that it still plays a notable political role because many journalists, political figures, and others with significant offline platforms heavily use the social media site. Because we only use the Decahose feed for our analysis, though, we cannot speak to the extent to which different demographics gave attention to victims of police violence and amplified their names. It would be beneficial to use a panel-based approach [70] to understand differences in how different racial, political, and other demographic groups engage with instances of police violence. Finally, Twitter is only one social media platform, and it would be valuable to understand the ways in which those on different platforms, like Facebook, Reddit, and TikTok, give attention to and amplify the names of Black victims of police violence.

## 6 Conclusion

Remembering Black victims of police violence as people, first and foremost, is key to understanding the effects of racism and enacting policy that addresses it. Here, we have shown that since the growth of the Black Lives Matter movement in 2014, people have regularly and increasingly memorialized those victims by saying their names. Moreover, we have shown how the quantitatively extraordinary salience of George Floyd's murder is one moment in a broader movement for racial justice and for the dignity of Black lives.

While a substantial number of Black people have been recognized as victims of fatal police violence, we emphasize that the majority of Black victims still receive little attention online. There are many reasons that is the case. We urge scholars of Twitter data and other social media data to further investigate not only those factors that lead some Black people subject to police violence to receive less online attention, but also the strategies employed by members of Black counterpublics to advocate for positive social change. We have attempted to contribute to this investigation here. In addition, and no less important, our study presents a moment to reflect on how we give attention to and amplify Black victims of police violence, and what inequities may still exist in how we do so. As the Black Lives Matter movement enters its eighth year of campaigning for racial justice, we all have an opportunity to ask how we can best memorialize and honor victims of police violence by saying their names.

## Supporting information

**S1 Appendix. Data preprocessing, mathematical definitions, word contributions to expressed happiness.**
(PDF)

**S1 Fig. Words that contributed to the "saddest" day recorded in Twitter's history, May 31st, 2020.** The week prior to George Floyd's death is used as a reference period.
(PDF)

**S2 Fig. Words that contributed to decreased happiness on May 26th, 2020, the day following George Floyd's death.** The week prior to George Floyd's death is used as a reference period.
(PDF)

**S3 Fig. Words that contributed to decreased happiness on May 29th, 2020.** The week prior to George Floyd's death is used as a reference period.
(PDF)

**S4 Fig. Words that contributed to decreased happiness on June 7th, 2020, the end of the spike period.** The week prior to George Floyd's death is used as a reference period.
(PDF)

**S5 Fig. Attention decay relative to the death of a victim and the day that their name received peak attention.** Scatter plots of normalized attention $\widehat{p_{\tau,t}}$ for all Black victims in the combined database anchored to the first 50 days after the date of death (left) and peak relative frequency (right). We exclude names that did not receive measurable attention during the specified period, i.e. did not appear in the top million 2-grams. The red line plot shows the mean of the included values on each day. Because of the log-scaled y-axis, points with a normalized frequency of 0 are not visible but are included in the mean.
(PDF)

**S6 Fig. Attention decay by gender.** See S5 Fig for details.
(PDF)

**S7 Fig. Histogram of delay between date of incident and date of peak rank.** The main figure uses 25-day bins, while the inset uses 1-day bins from 0 to 100 days between incident and peak rank. Short delays are more common than long delays, with 0 and 1 days between the incident and the peak rank of a name being the most common. This provides evidence that most names in our analyses are properly disambiguated.
(PDF)

**S8 Fig. Proportion of days with measurable attention and peak rank of Black male victims of fatal police violence.** See Fig 4 for details.
(PDF)

**S9 Fig. Proportion of days with measurable attention and peak rank of Black female victims of fatal police violence.** See Fig 4 for details.
(PDF)

**S10 Fig. Amplification of additional emblematic names of Black Lives Matter.** We see patterns similar to those observed in Fig 5, such as high relative social amplification ($> 1$) and dual peaks of attention.
(PDF)

**S11 Fig. Histogram of mean relative social amplification across names, separated by gender. The bin size is 0.005.** Although there are many more men than women in our analysis, their distributions of mean relative social amplification are similar. Many are concentrated around 1 and the distributions skew right, with more values above 1 than below it.
(PDF)

**S12 Fig. Histogram of maximum relative social amplification across names, separated by gender.** The bin size is 0.1. Although there are many more men than women in our analysis, their distributions of maximum relative social amplification are similar. The distributions

skew right. There are some outliers (not shown), all men, with a maximum relative social amplification above 10.
(PDF)

**S1 Table. Duplicate names in the Fatal Encounters database.** We attribute all mentions of a name to the earliest incident with a victim of that name, indicated by the date in the table.
(PDF)

**S2 Table. Names excluded because they received measurable attention in the ten days prior to death.** A person is said to have received measurable attention in the ten days prior to their death if their name was among the top million 2-grams in the ten days prior. This indicates that the name may be shared with another prominent figure, and so they excluded from the analysis. Note that "Walter Scott" was later added manually because the attention given to the victim of police violence greatly exceeded other usages of the name.
(PDF)

**S3 Table. Manually excluded names for disambiguation purposes.** The automatic name disambiguation process (see S2 Table) misses some cases where most uses of the name on Twitter clearly relate to someone other than the victim of police violence. We remove these manually.
(PDF)

**S4 Table. Cases where the name of the victim was withheld.** Some names in the Fatal Encounters database are "withheld by police"; therefore, we cannot determine their usage on Twitter. We list the date and location of these cases.
(PDF)

**S5 Table. Names that were manually added to the analysis.**
(PDF)

**S6 Table. Robustness of resurgent attention to past victims of police violence following George Floyd's death.** We consider four periods of spikes in attention relevant to #BlackLivesMatter: November 24th–December 8th, 2014 (deaths and non-indictments in the cases of Michael Brown, Tamir Rice, and Eric Garner), July 13th–July 26th, 2015 (death of Sandra Bland), July 5th–July 13th, 2016 (deaths of Philando Castile and Alton Sterling), August 12th–August 22nd, 2017 ("Unite the Right" Charlottesville rally), and May 25th–June 6th, 2020 (death of George Floyd). We vary the period before and after the spike across $n$ = 7, 60, and 90 days. The number of names that received increased attention during a spike period is reported, as well as the percentage of those that had not received any measurable attention in the $n$ days prior to the spike. The average change in average relative frequency is calculated for the difference between $n$ days before the spike period and during it, and $n$ days before and after it. Statistical significance is indicated by * for $\alpha$ = 0.05 and ** for $\alpha$ = 0.01.
(PDF)

**S7 Table. Names of those who received increased attention during the spike following George Floyd's death.** A name received increased attention if its the mean relative frequency from May 25 to June 7, 2020 was greater than its mean relative frequency from April 25 to May 24, 2020.
(PDF)

**S8 Table. Names of those who received increased attention during the spike following George Floyd's death.** A name received increased attention if its the mean relative frequency from May 25 to June 7, 2020 was greater than its mean relative frequency from April 25 to

May 24, 2020.
(PDF)

## Author Contributions

**Conceptualization:** Henry H. Wu, Brooke Foucault Welles, Christopher M. Danforth.

**Data curation:** Henry H. Wu, Thayer Alshaabi, Jane L. Adams, Joshua R. Minot, Michael V. Arnold, Peter Sheridan Dodds, Christopher M. Danforth.

**Formal analysis:** Henry H. Wu, Brooke Foucault Welles, Randall Harp.

**Funding acquisition:** Peter Sheridan Dodds, Christopher M. Danforth.

**Investigation:** Henry H. Wu, Ryan J. Gallagher, Thayer Alshaabi, Brooke Foucault Welles, Randall Harp, Christopher M. Danforth.

**Methodology:** Henry H. Wu, Ryan J. Gallagher, Thayer Alshaabi, Michael V. Arnold, Brooke Foucault Welles, Peter Sheridan Dodds, Christopher M. Danforth.

**Project administration:** Christopher M. Danforth.

**Resources:** Joshua R. Minot.

**Software:** Henry H. Wu, Thayer Alshaabi, Jane L. Adams, Joshua R. Minot, Michael V. Arnold, Peter Sheridan Dodds.

**Supervision:** Brooke Foucault Welles, Randall Harp, Christopher M. Danforth.

**Visualization:** Henry H. Wu, Ryan J. Gallagher, Thayer Alshaabi, Jane L. Adams, Joshua R. Minot, Michael V. Arnold, Peter Sheridan Dodds.

**Writing – original draft:** Henry H. Wu, Ryan J. Gallagher.

**Writing – review & editing:** Ryan J. Gallagher, Thayer Alshaabi, Jane L. Adams, Joshua R. Minot, Michael V. Arnold, Brooke Foucault Welles, Randall Harp, Peter Sheridan Dodds, Christopher M. Danforth.

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
