## [Decision Letter · Decision Letter 0]

26 Aug 2022

PONE-D-22-19806Say Their Names: Resurgence in the collective attention toward Black victims of fatal police violence following the death of George FloydPLOS ONE

Dear Dr. Danforth,

Thank you for submitting your manuscript to PLOS ONE. After careful consideration, we feel that it has merit but does not fully meet PLOS ONE’s publication criteria as it currently stands. Therefore, we invite you to submit a revised version of the manuscript that addresses the points raised during the review process. 

The three reviewers have considerable expertise in the substantive and methodological issues under investigation.

Reviewer 1 recommended acceptance but also offered a couple of suggestions. The first refers to the gender of the victims of police killings. The second refers to the broader significance of the study (an issue also raised by reviewer 2). 

Reviewer 2 recommended minor revisions and outlined some areas for improvement. Like reviewer 1 they would like to hear more of your thoughts about the broader significance of this specific case study. Was George Floyd—and the social movement it energised—so unusual in quality and quantity that its singularity means that one cannot really generalize to past, current and future instances of social justice movements and their political context? If so, why? If not, what broader insights can be learnt? Referee 2 also offers some thoughts on the distinction between online and offline activism. 

Reviewer 3, who like the other two referees is also exceptionally well placed to assess the paper, recommended minor revisions, and made a few points of a methodological/analytical nature. The first refers to quantifying attention relative to appropriate comparison points. The second relates to why some names may have received more attention (and had different temporal dynamics in terms of attention) than other names. The third is about the readability of Figure 1.

On the basis of these reviews, I am asking for minor revisions . This is an interesting paper and I look forward to reading the revised version. You do not necessarily need to redraft the manuscript in response to each and every comment, but please do detail in a ‘response to review’ document your thoughts about the points made by the referees.

We look forward to receiving your revised manuscript.

Kind regards,

Jonathan Jackson, Ph.D

Academic Editor

PLOS ONE

Journal Requirements:

2. Please ensure that you include a statement specifying whether the collection and analysis method complied with the terms and conditions of the data source.

"PSD and CMD received support from a gift to the UVM Foundation from MassMutual"

Reviewers' comments:

Reviewer's Responses to Questions

**Comments to the Author**

1. Is the manuscript technically sound, and do the data support the conclusions?

Reviewer #1: Yes

Reviewer #2: Yes

Reviewer #3: Yes

2. Has the statistical analysis been performed appropriately and rigorously? 

Reviewer #1: Yes

Reviewer #2: Yes

Reviewer #3: Yes

3. Have the authors made all data underlying the findings in their manuscript fully available?

Reviewer #1: Yes

Reviewer #2: Yes

Reviewer #3: Yes

4. Is the manuscript presented in an intelligible fashion and written in standard English?

Reviewer #1: Yes

Reviewer #2: Yes

Reviewer #3: Yes

5. Review Comments to the Author

Reviewer #1: This essay could benefit from a bit more gender analysis. I appreciate the authors explaining the origins of Say their Names, derived from #SayHerName, but I wanted to see a bit more about how the number of victims remembered and called forth after George Floyd's murder were men. I would like to understand the significance of the saddest day on Twitter, i.e. what new precedents and considerations might be helpful for the platform to develop or attend to sadness should there be another day that is equally sad or sadder. The conclusion could benefit with a more declaritive call to action for scholars of Twitter data and some more clarity about the significance of these findings and to whom. It's a great paper and really doesn't need much. There is a typo in the abstract with "ever" repeated.

Reviewer #2: Overall, this is an interesting study. I think it clearly contributes to our knowledge of Black Lives Matter (BLM) in two primary ways. While other scholars have published data showing spikes in attention around BLM after the George Floyd murder, this piece is exceptional (to my knowledge) in specifically studying sentiment on Twitter around Floyd's death and the ways Floyd's name was connected with 185 other individuals as a result of 'say their names' as a discursive strategy. The authors also provide a smart conceptualization of this as a rhetorical/narrative strategy that connects this murder to others, and in turn, creates a longer history of witnessing of police violence.

For this reason, I think this piece makes a clear and important empirical contribution to our understanding of this particular movement and moment. I am less clear on its theoretical contribution - or how much we can logically generalize beyond this movement, this strategy, this struggle for racial justice, to understand movement tactics more generally, or even conceptualize this particular movement. In other words, where are the analytical or empirical limits of this case. The paper seems to imply that the rhetorical and narrative tactic of say their names, in being grounded in a particular Black tradition of witnessing, is in fact unique to this context - but I think the paper would be well served to come out and say that then explicitly as part of the case conceptualization. Indeed, there are extensive citations to Richardson's book here, but not for its singular theoretical achievement - namely, that scholars need to abandon any generalized theory of witnessing shorn of race, ethnicity, context, history, and power. Alternatively, if that is not the argument (i.e.: that a Black struggle for racial justice is, in fact, singular), the authors would be well served to step outside of the narrowness of their case here and logically generalize to other movements for either racial justice, or other cases of hashtag activism; theoretically, would rhetorical or narrative strategies like this also further movements for justice in other contexts? If not, why not? Why don't we see similar tactics in other movements? Or, if we do, why are they presumably not as successful?

This matters, because as it stands this really is solely a descriptive study of a prominent (if not the prominent) racial justice movement. I think it is an important contribution in its own right - that is not at issue. But the piece would be considerably strengthened by at least thinking about this case as a case - what are its connections to other cases, or limits (in the Richardson view) given its singular history?

Empirically, I think the use of the Fatal Encounters database is novel and very interesting. The empirical results around attention and amplification are pretty extraordinary and clearly empirically document what at least seems to be an overarching aim of the movement (and confirm Richardson's broad historical continuity framework). I think the limitations section here is well done - I had similar questions about the use of the Fatal Encounters database (but also know enough about the lack of standardization in police reporting to see its use as being well-justified.) I agree that data on knowing why these names, and not others, would be important. I also agree that there is an open question beyond the descriptive here about just how effective this is as a tactic - how is it connected to wider policy outcomes or power outcomes? Does it set the agenda across other mediums, or in Democratic Party policy-making and elite rhetoric? I think much more could be done to at least discuss these possibilities (drawing in other work on how social movement activism works in tandem with social media and how there is a diverse range of targets and outcomes). I would also drop any notion of online vs. offline activism, which sociologists have long argued against. It also blunts the force of the findings here - movements are always online and offline simultaneously, there are no meaningful distinctions, only strategics, tactics, tools, and goals that work in tandem.

Reviewer #3: This article uses data from Twitter to demonstrate that the murder of George Floyd by police in 2020 led to an unprecedented increase in the levels of public attention to the Black Lives Matter movement, which is illustrated specifically by spikes in mentions of the names of the other Black victims of police violence.

This is a strong manuscript that offers detailed and novel descriptive evidence about the dynamics of attention and amplification of content related to the Black Lives Matter movement on Twitter.

While other papers or policy briefs (most notably the work by Deen Freelon) have already offered descriptive evidence about the role that social media played in the spread of the the Black Lives Matter movement, my view is that this article represents a major contribution to the literature in three different ways. First, to my knowledge the longitudinal analysis presented here is based on the longest time period analyzed in any previous study. Second, it develops novel new quantitative metrics of attention and amplification that help track how specific words become salient during particular points in time, and how they are spread via retweets, and which I foresee will be used in future work. Finally, the article relies on excellent data visualizations that elegantly display time trends and which, again, I see as a notable methodological contribution to the literature.

Despite these strengths, the article also has some shortcoming that I believe should be addressed before it can be published.

First, the article could do more to contextualize the time trends that are described, and how attention to Black Lives Matter and the names of Black victims of police violence compares to other topics. I understand that many of the metrics are normalized with respect to the most frequent words, but I think more could be done to more clearly illustrate why this level of attention is unprecedented. In page 10 for example we get a few examples about the attention peaks in comparison to other proper names. But what about other common bigrams, such as those related to news events, TV shows, pop stars, etc.?

Second, by the end of the paper I was somewhat missing a more thorough attempt to describe which names received more attention and which names did not. The authors appear to leave this to future work (see pp.19-20), but I wonder if more could be done here to at least anecdotally describe any initial patterns that emerge from the data, so that future researchers can at least have some priors to build on. Similarly, I would have loved to see more descriptive evidence not only on the spikes, but also on how long attention persists and then decays, and whether it decays at a similar pace for all names.

Finally, I would encourage the authors to revisit Figure 1, in which they describe the data. It was extremely hard to read, particularly the top facet. The y-axis labels could also be improved by adding more informative labels. I would recommend increasing its size. Also, why does it only start on 2020, compared to other graphs that start much earlier?

6. PLOS authors have the option to publish the peer review history of their article (what does this mean?). If published, this will include your full peer review and any attached files.

Reviewer #1: No

Reviewer #2: No

Reviewer #3: No

---

## [Editor Report · Decision Letter 1]

5 Dec 2022

Say Their Names: Resurgence in the collective attention toward Black victims of fatal police violence following the death of George Floyd

PONE-D-22-19806R1

Dear Dr. Danforth,

We’re pleased to inform you that your manuscript has been judged scientifically suitable for publication and will be formally accepted for publication once it meets all outstanding technical requirements.

Kind regards,

Jonathan Jackson, Ph.D

Academic Editor

PLOS ONE
---

## [Editor Report · Acceptance letter]

20 Dec 2022

PONE-D-22-19806R1 

Say Their Names: Resurgence in the collective attention toward Black victims of fatal police violence following the death of George Floyd 

Dear Dr. Danforth:

I'm pleased to inform you that your manuscript has been deemed suitable for publication in PLOS ONE. Congratulations! Your manuscript is now with our production department. 

Kind regards, 

on behalf of

Dr. Jonathan Jackson 

Academic Editor

PLOS ONE